# MIBench: A Comprehensive Benchmark for Model Inversion Attack and Defense

## Abstract

Model Inversion (MI) attacks aim at leveraging the output information of target models to reconstruct privacy-sensitive training data, raising widespread concerns on privacy threats of Deep Neural Networks (DNNs). Unfortunately, in tandem with the rapid evolution of MI attacks, the lack of a comprehensive, aligned, and reliable benchmark has emerged as a formidable challenge. This deficiency leads to inadequate comparisons between different attack methods and inconsistent experimental setups. In this paper, we introduce the first practical benchmark for model inversion attacks and defenses to address this critical gap, which is named *MIBench*. This benchmark serves as an extensible and reproducible modular-based toolbox and currently integrates a total of 16 state-of-the-art attack and defense methods. Moreover, we furnish a suite of assessment tools encompassing 9 commonly used evaluation protocols to facilitate standardized and fair evaluation and analysis. Capitalizing on this foundation, we conduct extensive experiments from multiple perspectives to holistically compare and analyze the performance of various methods across different scenarios, which overcomes the misalignment issues and discrepancy prevalent in previous works. Based on the collected attack methods and defense strategies, we analyze the impact of target resolution, defense robustness, model predictive power, model architectures, transferability and loss function. Our hope is that this *MIBench* could provide a unified, practical and extensible toolbox and is widely utilized by researchers in the field to rigorously test and compare their novel methods, ensuring equitable evaluations and thereby propelling further advancements in the future development.

## 1 Introduction

In recent years, Model Inversion (MI) attacks have raised alarms over the potential privacy breaches of sensitive personal information, including the leakage of privacy images in face recognition models (He et al., 2016), sensitive health details in medical data (Wang et al., 2022), financial information such as transaction records and account balances (Ozbayoglu et al., 2020), and personal preferences and social connections in social media data (Feng et al., 2022). In the MI attacks, an attacker aims to infer private training data from the output information of the target model. Fredrikson et al. (2014) proposed the first MI attack against linear regression models to reconstruct sensitive features of genomic data. Subsequent studies (Fredrikson et al., 2015; Song et al., 2017; Yang et al., 2019) have extended MI attacks to more Machine Learning (ML) models. Zhang et al. (2020b) first introduces the GANs as stronger image priors, paving the way for more applications in GAN-based methods (Chen et al., 2021a; Yuan et al., 2022; 2023b; Wang et al., 2021b).

However, some critical challenges are posed owing to the lack of a comprehensive, fair, and reliable benchmark. Specifically, the evaluation of new methods is often confined to comparisons with a narrow selection of prior works, limiting the scope and depth of analysis. For instance, some methods (Nguyen et al., 2023b; Yuan et al., 2023a) exhibit superior performance for lower-resolution images while other methods (Struppek et al., 2022; Qiu et al., 2024) perform better at higher resolutions. However, these studies only evaluate under their predominant resolutions, and thus do not provide a unified and comprehensive comparison. Additionally, the absence of unified experimental protocols results in a fragmented landscape where there is less validity and fairness in the comparative studies. For example, the original GMI (Zhang et al., 2020a) achieves the attack accuracy of 28%, 44%, 46% when attacking three different classifiers trained on the CelebFaces Attributes(CelebA) (Liu

et al., 2015), while merely maintaining lower attack accuracy of 21%, 32%, 31% and 21%, 31%, 29% when compared in the KEDMI (Chen et al., 2021b) and PLGMI (Yuan et al., 2023a) under the same experimental setup. Compounding these issues, discrepancies in the evaluation metrics further obscure the reliability of reported conclusions, potentially steering the field towards biased or misleading insights. These shortcomings impede both the accurate measurement of advancements in the MI field and the systematic exploration of its theoretical underpinnings, underscoring the urgent need for a harmonized framework to facilitate robust and transparent research practices.

To alleviate these problems, we establish the first benchmark in MI field, named *MIBench*. For building an extensible modular-based toolbox, we disassemble the pipeline of MI attacks and defenses into four main modules, each designated for data preprocessing, attack methods, defense strategies and evaluation, hence enhancing the extensibility of this unified framework. The proposed *MIBench* has encompassed a total of 16 distinct attack and defense methods, coupled with 9 prevalent evaluation protocols to adequately measure the comprehensive performance of individual MI methods. Furthermore, we conduct extensive experiments from diverse perspectives to achieve a thorough appraisal of the competence of existing MI methods, while simultaneously venturing into undiscovered insights to inspire potential avenues for future research. We expect that this reproducible benchmark will facilitate the further development of MI field and bring more innovative explorations in the subsequent study. Our main contributions are as follows:

- We build the first comprehensive benchmark in MI field, which serves as an extensible and reproducible modular-based toolbox for researchers. We expect that this will foster the development of more powerful MI attacks by making it easier to evaluate their effectiveness across multiple distinct dimensions.

- We implement 16 state-of-the-art attack methods and defense strategies and 9 evaluation protocols currently in our benchmark. We hope that the benchmark can further identify the most successful ideas in defending against rapid development of potential MI attacks.

- We conduct extensive experiments to thoroughly assess different MI methods under multiple settings and study the effects of different factors to offer new insights on the MI field. In particular, we validate that stronger model predictive power correlates with an increased likelihood of privacy leakage. Moreover, our analysis reveals that certain defense algorithms also fail when the target model achieves high prediction accuracy.

## 2  RELATED WORK

**Model Inversion Attacks.** In the MI attacks, the malicious adversary aims to reconstruct privacy-sensitive data by leveraging the output prediction confidence of the target classifier and other auxiliary priors. Normally, the attacker requires a public dataset that shares structural similarities with the private dataset but without intersecting classes to pre-train the generator. For example, an open-source face dataset serves as essential public data when targeting a face recognition model. For a typical GAN-based MI attack, attackers attempt to recover private images $\mathbf{x}^*$ from the GAN's latent vectors $\mathbf{z}$ initialized by Gaussian distribution, given the target image classifier $f_\theta$ parameterized with weights $\theta$ and the trained generator $G$. The attack process can be formulated as follows:

$$\mathbf{z}^* = \arg\min_{\mathbf{z}} \mathcal{L}_{id}(f_\theta(G(\mathbf{z})), c) + \lambda \mathcal{L}_{aux}(\mathbf{z}; G), \tag{1}$$

where $c$ is the target class, $\mathcal{L}_{id}(\cdot, \cdot)$ typically denotes the classification loss, $\lambda$ is a hyperparameter, and $\mathcal{L}_{aux}(\cdot)$ is the prior knowledge regularization (e.g., the discriminator's classification loss) used to improve the reality of $G(\mathbf{z})$. By minimizing the above equation, the adversary updates the latent vectors $\mathbf{z}$ into the optimal results $\hat{\mathbf{z}}$ and generate final images through $\hat{\mathbf{x}} = G(\hat{\mathbf{z}})$.

Fredrikson et al. (2014) first introduce the concept of MI attacks in the context of genomic privacy. They find that maximizing the posterior probabilities of a linear regression model can reconstruct the original genomic markers. Ensuing works (Fredrikson et al., 2015; Song et al., 2017; Yang et al., 2019) manage to design MI attacks for more kinds of models and private data, but are still limited to attacking simple networks and grayscale images. To enhance the reconstruction performance on complex RGB images, GMI (Zhang et al., 2020a) first propose to incorporate the rich prior knowledge (Fang et al., 2023; Gu et al., 2020; Fang et al., 2024) within the pre-trained Generative Adversarial Networks (GANs) (Goodfellow et al., 2014). Specifically, GMI starts by generating a

series of preliminary fake images, and then iteratively optimizes the input latent vectors that are used for generation. Based on GMI, KEDMI (Chen et al., 2021b) refine the discriminator by introducing target labels to recover the distribution of the input latent vectors. VMI (Wang et al., 2021a) utilize variational inference to model MI attacks and adopts KL-divergence as the regularization to better estimate the target distribution. PPA (Struppek et al., 2022) introduce a series of techniques such as initial selection, post-selection, and data argumentation to enhance MI attacks and manages to recover high-resolution images by the pre-trained StyleGAN2 (Karras et al., 2019). LOMMA (Nguyen et al., 2023b) integrate model augmentation and model distillation into MI attacks to tackle the problem of over-fitting. PLGMI (Yuan et al., 2023a) leverage a top-$n$ selection technique to generate pseudo labels to further guide the training process of GAN.

Besides, based on whether the parameters and structures of the victim model are fully accessible to the attackers, MI attacks can be further split into *white-box* attacks and *black-box* attacks. Note that in black-box settings, the gradients can no longer be computed by the back-propagation process. Thus, Yuan et al. (2022) address this problem by sampling numerous latent vectors from random noise, selecting the ones that can generate correct labels, and updating the latent vectors solely on discriminator loss. Nguyen et al. (2023a) propose to conduct MI attacks on various surrogate models instead of the unfamiliar victim model, transforming the black-box settings into white-box settings.

**Model Inversion Defenses.** To defend MI attacks, most existing methods can be categorized into two types: *model output processing* (Yang et al., 2020; Wen et al., 2021; Ye et al., 2022) and *robust model training* (Gong et al., 2023; Titcombe et al., 2021; Li et al., 2022; Wang et al., 2021c; Peng et al., 2022; Struppek et al., 2023). Model output processing refers to reducing the private information carried in the victim model's output to promote privacy. Yang et al. (2020) propose to train an autoencoder to purify the output vector by decreasing its degree of dispersion. Wen et al. (2021) apply adversarial noises to the model output and confuse the attackers. Ye et al. (2022) leverage a differential privacy mechanism to divide the output vector into multiple sub-ranges. Robust model training refers to that incorporating the defense strategies during the training process. MID Wang et al. (2021c) penalizes the mutual information between model inputs and outputs in the training loss, thus reducing the redundant information carried in the model output that may be abused by the attackers. However, simply decreasing the dependency between the inputs and outputs also results in model performance degradation. To alleviate this issue and strike a better balance between model utility and user privacy, Bilateral Dependency Optimization (BiDO) (Peng et al., 2022) minimizes the dependency between the inputs and outputs while maximizing the dependency between the latent representations and outputs. Gong et al. (2023) propose to leverage GAN to generate fake public samples to mislead the attackers. Titcombe et al. (2021) defend MI attacks by adding Laplacian noise to intermediate representations. LS (Struppek et al., 2023) finds that label smoothing with negative factors can help privacy preservation. TL (Ho et al., 2024) leverages transfer learning to limit the number of layers encoding sensitive information and thus improves the robustness to MI attacks.

## 3 OUR BENCHMARK

### 3.1 DATASET

Considering existing MI attacks primarily focus on reconstructing private facial data from image classifiers, we select 4 widely recognized face datasets as the basic datasets for our benchmark, which include Flickr-Faces-HQ (FFHQ) (Karras et al., 2019), MetFaces (Karras et al., 2020a), FaceScrub (Ng & Winkler, 2014), and CelebFaces Attributes (CelebA) (Liu et al., 2015). Generally, FFHQ and MetFaces are employed as public datasets for pre-training auxiliary priors, whereas FaceScrub and CelebA serve as the target private datasets to attack. Our benchmark facilitates researchers to freely combine public datasets with private ones, thereby enabling customized experimental setups. Extensive evaluation on more non-facial datasets are presented in Sec.C.3

Notably, the target resolutions across different MI attacks are not uniform. The majority of attack methods concentrate on low-resolution images of $64 \times 64$, while recent attack methods have begun to focus on higher resolutions, such as $224 \times 224$. Therefore, our benchmark offers 2 versions of low-resolution and higher-resolution for the aforementioned 4 datasets and prepares multiple transformation tools for processing images, freeing researchers from the laborious tasks of data preprocessing. More details regarding the datasets can be found in the Sec. B.2 of the Appendix.

## 3.2 Implemented Methods

Our benchmark includes a total of 16 methods, comprising 11 attack methods and 4 defense strategies. With a focus on Generative Adversarial Network (GAN)-based MI attacks, we selectively reproduce methods from recent years that have been published in top-tier conferences or journals in the computer vision or machine learning domains. This criterion ensures the reliability and validity of the implemented methods. Considering the main targets in our benchmark are image classifiers for RGB images, the learning-based MI attacks (Fredrikson et al., 2015; Song et al., 2017; Yang et al., 2019) are not incorporated currently. More detailed information about the implemented methods is stated in Sec. A.2 and Sec. A.3.

***Attacks.*** Based on the accessibility to the target model's parameters, we categorize MI attacks into *white-box* and *black-box* attacks. *White-box* attacks can entail full knowledge of the target model, enabling the computation of gradients for performing backpropagation, while *black-box* attacks are constrained to merely obtaining the prediction confidence vectors of the target model. Our benchmark includes 8 *white-box* attack methods and 4 *black-box* attack methods, as summarized in Table 1.

Table 1: Summary of implemented MI attack methods in our benchmark.

| Attack Method | Accessibility | Reference | GAN Prior | Official Resolution |
|---|---|---|---|---|
| GMI (Zhang et al., 2020a) | *White-box* | CVPR-2020 | WGAN-GP | $64 \times 64$ |
| KEDMI (Chen et al., 2021b) | *White-box* | ICCV-2021 | Inversion-specific GAN[†] | $64 \times 64$ |
| VMI (Wang et al., 2021a) | *White-box* | NeurIPS-2021 | StyleGAN2 | $64 \times 64$ |
| Mirror* (An et al., 2022) | *White-box/Black-box* | NDSS-2022 | StyleGAN | $224 \times 224$ |
| PPA (Struppek et al., 2022) | *White-box* | ICML-2022 | StyleGAN2 | $224 \times 224$ |
| PLGMI (Yuan et al., 2023a) | *White-box* | AAAI-2023 | Conditional GAN | $64 \times 64$ |
| LOMMA (Nguyen et al., 2023b) | *White-box* | CVPR-2023 | ~ | $64 \times 64$ |
| IF-GMI (Qiu et al., 2024) | *White-box* | ECCV-2024 | StyleGAN2 | $224 \times 224$ |
| BREPMI (Kahla et al., 2022) | *Black-box* | CVPR-2022 | WGAN-GP | $64 \times 64$ |
| C2FMI (Ye et al., 2023) | *Black-box* | TDSC-2023 | StyleGAN2 | $160 \times 160$ |
| RLBMI (Han et al., 2023) | *Black-box* | CVPR-2023 | WGAN-GP | $64 \times 64$ |
| LOKT (Nguyen et al., 2023a) | *Black-box* | NeurIPS-2023 | ACGAN | $128 \times 128$ |

*Mirror (An et al., 2022) proposes attack methods on both *white-box* and *black-box* settings.
[†]KEDMI (Chen et al., 2021b) first proposed this customized GAN.
~LOMMA (Nguyen et al., 2023b) is a plug-and-play technique applied in combination with other MI attacks.

***Defenses.*** To effectively defend against MI attacks, the defender typically employs defense strategies during the training process of victim classifiers. Our benchmark includes 4 typical defense strategies and the details are presented in Table 2.

Table 2: Summary of implemented MI defense strategies in our benchmark.

| Defense Strategy | Reference | Core Technique | Description |
|---|---|---|---|
| MID (Wang et al., 2021c) | AAAI-2021 | Regularization | Utilize mutual information regularization to limit leaked information about the model input in the model output |
| BiDO (Peng et al., 2022) | KDD-2022 | Regularization | Minimize dependency between latent vectors and the model input while maximizing dependency between latent vectors and the outputs |
| LS (Struppek et al., 2023) | ICLR-2024 | Label Smoothing | Adjusting the label smoothing with negative factors contributes to increasing privacy protection |
| TL (Ho et al., 2024) | CVPR-2024 | Transfer Learning | Utilize transfer learning to limit the number of layers encoding privacy-sensitive information for robustness to MI attacks |

## 3.3 Toolbox

We implement an extensible and reproducible modular-based toolbox for our benchmark, as shown in Fig 1. The framework can be divided into four main modules, including *Data Preprocess Module*, *Attack Module*, *Defense Module* and *Evaluation Module*.

***Data Preprocess Module.*** This module is designed to preprocess all data resources required before launching attacks or defenses, including datasets, classifiers and parameters. Consequently, we furnish this module with three fundamental functionalities: *dataset preprocessing*, *target model*

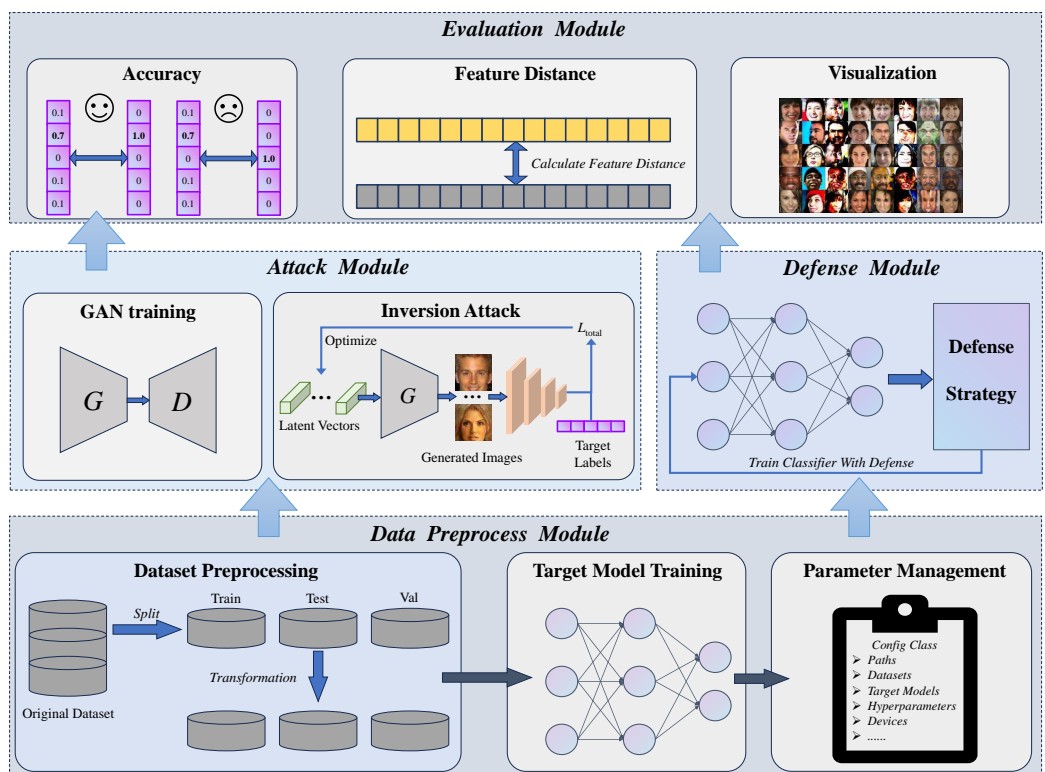

Figure 1: Overview of the basic structure of modular-based toolbox for our benchmark.

*training*, and *parameter management*. For **dataset preprocessing**, we build a unified pipeline for each dataset, which automatically carries out a series of operations such as spliting dataset and image transformations (*e.g.* center crop and resize) based on the split file and chosen resolution from users. For **target model training**, users can further leverage the processed datasets to train designated classifiers. We abstract the general procedures in classifier training into a base trainer class to facilitate users in extending and customizing their own classifiers. For **parameter management**, we encapsulate parameters used in different processes into specific configuration classes, such as TrainConfig designated for the training process and AttackConfig for the attack process, thus ensuring organized and efficient parameter handling across various operations in the workflow.

*Attack Module.* The workflow of MI attacks can be roughly divided into two stages. The first stage is **GAN training**, where the module abstracts the general training process of GANs into a basic trainer class. This allows users not only train GANs that are pre-built into the benchmark, but also extend to their uniquely designed GANs. The second stage is the core **inversion attack**, which we split into three parts: *latent vectors initialization*, *iterative optimization*, and an optional *post-processing* step. After completion of the attack, the module preserves essential data such as the final optimized images and latent vectors, facilitating subsequent evaluation and analysis.

*Defense Module.* Considering the mainstream MI defense strategies are applied during the training process of target classifiers, we design the defense module following the target model training functionality within the *Data Preprocessing Module*. To enhance extensibility, we incorporate defense strategies as part of the training parameters for classifiers to enable the defense during the training process of target models, which decouples the defense from the training pipeline. In this way, we allow users to customize their own defense strategies against MI attacks.

*Evaluation Module.* Our benchmark concentrates on the evaluation at distribution level instead of sample level, assessing the overall performance of the whole reconstructed dataset. Therefore, we provide a total of 9 widely recognized distribution level evaluation metrics for users, which can be categorized into four types according to the evaluated content: *Accuracy*, *Feature Distance*,

*FID* (Heusel et al., 2017) and *Sample Diversity*. For **Accuracy**, this metric measures how well the reconstructed samples resemble the target class, consisting of **Acc@1** and **Acc@5**. For **Feature Distance**, it computes the average shortest $l_2$ distance from features between each reconstructed sample and private data to measure the similarity between the feature space of reconstructed samples and private dataset. For **FID**, the lower FID score shows higher realism and overall diversity (Wang et al., 2021a). For **Sample Diversity**, the higher values indicate greater intra-class diversity. Moreover, we provide convenient tools for analysis, including *Standard Deviation Calculation* and *Visualization*.

# 4 EXPERIMENT

## 4.1 EXPERIMENTAL SETUPS

To ensure fair and uniform comparison and evaluation, we select the FFHQ (Karras et al., 2019) as the public dataset and FaceScrub (Ng & Winkler, 2014) as the private dataset for all the experiments in the Experiment section. The target models are fixed to the IR-152 (He et al., 2016) for low-resolution scenario and ResNet-152 (He et al., 2016) for high-resolution scenario, both trained on the FaceScrub. For each attack method, the number of images reconstructed per class is set to 5 due to considerations of time and computation cost. More detailed experimental settings are listed in Sec.B in the Appendix.

Notably, we limit the evaluation exhibited in the Experiment section to merely three metrics, including *Accuracy*, *Feature Distance* and *FID* (Heusel et al., 2017), while *Sample Diversity* is presented in the Sec.C.1 in the Appendix. Moreover, the VMI (Wang et al., 2021a) and RLBMI (Han et al., 2023) will be further evaluated in Sec. C.7 owing to their excessive need of time.

## 4.2 EVALUATION ON DIFFERENT ATTACK METHODS

In this part, we prepare a unified experimental setting for different MI attack methods to conduct a fair comparison. The resolution of private and public datasets is set to $64 \times 64$, indicating a relatively easier scenario. Comparisons of white-box and black-box MI attacks are presented in Table 3.

Remarkably, the PLGMI (Yuan et al., 2023a) and LOKT (Nguyen et al., 2023a) achieve state-of-the-art comprehensive performance in white-box attacks and black-box attacks respectively, showing significant superiority in *Accuracy* and *Feature Distance* metrics. However, the lowest *FID* scores occur in the PPA (Struppek et al., 2022) and C2FMI (Ye et al., 2023) respectively instead of the above methods. We infer that this is because PPA and C2FMI employ more powerful generators (*e.g.* StyleGAN2 (Karras et al., 2020b)) as the GAN prior compared to PLGMI and LOKT, leading to more real image generation. Visualization in Fig 2 further validates the inference.

Table 3: Comparison between different white-box and black-box MI attacks.

| Method | ↑ **Acc@1** | ↑ **Acc@5** | ↓ $\delta_{eval}$ | ↓ $\delta_{face}$ | ↓ **FID** |
|---|---|---|---|---|---|
| GMI | $0.153 \pm 0.077$ | $0.265 \pm 0.093$ | $2442.667 \pm 298.597$ | $1.300 \pm 0.176$ | 91.861 |
| KEDMI | $0.404 \pm 0.017$ | $0.579 \pm 0.013$ | $2113.473 \pm 545.085$ | $0.997 \pm 0.337$ | 61.035 |
| Mirror(white) | $0.311 \pm 0.014$ | $0.509 \pm 0.021$ | $1979.211 \pm 427.343$ | $0.996 \pm 0.258$ | 36.610 |
| PPA | $0.844 \pm 0.036$ | $0.923 \pm 0.026$ | $1374.967 \pm 387.380$ | $0.657 \pm 0.195$ | **31.433** |
| PLGMI | $\mathbf{0.998} \pm 0.002$ | $\mathbf{0.999} \pm 0.001$ | $\mathbf{967.295} \pm 222.725$ | $\mathbf{0.486} \pm 0.103$ | 74.155 |
| LOMMA+GMI | $0.557 \pm 0.111$ | $0.678 \pm 0.096$ | $1948.976 \pm 317.310$ | $0.949 \pm 0.221$ | 62.050 |
| LOMMA+KEDMI | $0.711 \pm 0.007$ | $0.860 \pm 0.006$ | $1685.514 \pm 486.419$ | $0.759 \pm 0.289$ | 62.465 |
| IF-GMI | $0.797 \pm 0.018$ | $0.865 \pm 0.014$ | $1462.914 \pm 486.419$ | $0.722 \pm 0.232$ | 33.057 |
| BREPMI | $0.354 \pm 0.013$ | $0.608 \pm 0.015$ | $2178.587 \pm 357.194$ | $0.971 \pm 0.186$ | 74.519 |
| Mirror(black) | $0.526 \pm 0.031$ | $0.729 \pm 0.020$ | $1972.175 \pm 427.391$ | $0.854 \pm 0.239$ | 54.231 |
| C2FMI | $0.263 \pm 0.009$ | $0.459 \pm 0.016$ | $2061.995 \pm 534.556$ | $1.011 \pm 0.265$ | **43.488** |
| LOKT | $\mathbf{0.834} \pm 0.010$ | $\mathbf{0.918} \pm 0.013$ | $\mathbf{1533.071} \pm 402.791$ | $\mathbf{0.694} \pm 0.169$ | 71.701 |

## 4.3 EVALUATION ON HIGHER RESOLUTION

Recent attack methods have attempted to conquer higher resolution scenarios, such as PPA and Mirror. Accordingly, we conduct a further assessment of MI attacks under an increased resolution of

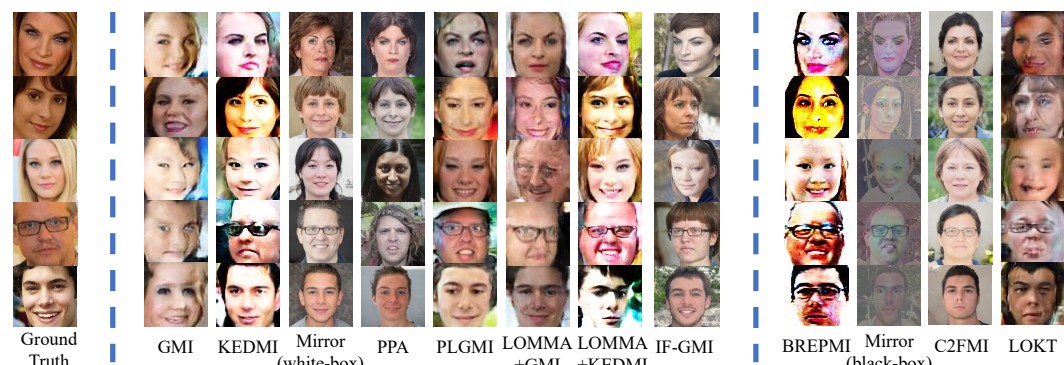

Figure 2: Visual comparison between different MI attacks.

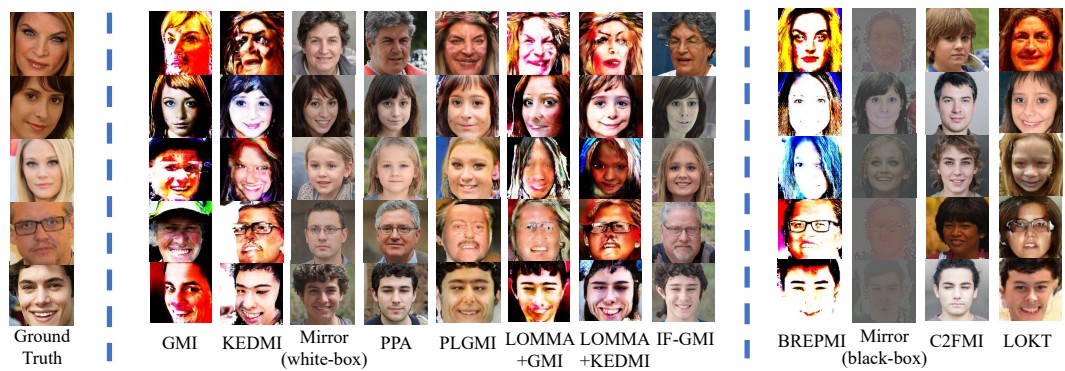

Figure 3: Visual comparison between different MI attacks on higher resolution scenario.

$224 \times 224$, which is considered a more challenging task. The evaluation results for white-box and black-box attacks are demonstrated in Table 4.

The results imply the significant impact of GAN priors when attacking private images with higher resolution. All the methods that employ stronger GAN priors maintain low FID scores, including Mirror, C2FMI, and PPA, while other methods suffer from significant degradation in image reality. This phenomenon is more pronounced in the visualization results displayed in Fig 3. Despite the primary metric for evaluating MI attacks is *Accuracy*, the reality of reconstructed images should be ensured within a reasonable range for better image quality. Thus, it is imperative to explore more complex GAN priors with enhanced performance in future research, extending the MI field to more challenging and practical applications.

Table 4: Comparison between white-box MI attacks on higher resolution scenario.

| Method | ↑ **Acc@1** | ↑ **Acc@5** | ↓ $\delta_{eval}$ | ↓ $\delta_{face}$ | ↓ **FID** |
|---|---|---|---|---|---|
| GMI | $0.073 \pm 0.024$ | $0.192 \pm 0.056$ | $\mathbf{134.640} \pm 24.203$ | $1.328 \pm 0.135$ | $119.755$ |
| KEDMI | $0.252 \pm 0.007$ | $0.494 \pm 0.013$ | $144.139 \pm 33.673$ | $1.139 \pm 0.214$ | $124.526$ |
| Mirror(white) | $0.348 \pm 0.023$ | $0.649 \pm 0.016$ | $197.741 \pm 32.212$ | $1.049 \pm 0.154$ | $59.628$ |
| PPA | $0.913 \pm 0.022$ | $0.986 \pm 0.004$ | $167.532 \pm 28.944$ | $0.774 \pm 0.143$ | $\mathbf{46.246}$ |
| PLGMI | $\mathbf{0.926} \pm 0.007$ | $\mathbf{0.987} \pm 0.002$ | $135.557 \pm 36.500$ | $0.730 \pm 0.177$ | $117.850$ |
| LOMMA+GMI | $0.735 \pm 0.043$ | $0.875 \pm 0.037$ | $136.700 \pm 29.743$ | $0.953 \pm 0.171$ | $111.151$ |
| LOMMA+KEDMI | $0.627 \pm 0.009$ | $0.864 \pm 0.006$ | $146.612 \pm 42.594$ | $0.977 \pm 0.244$ | $103.479$ |
| IF-GMI | $0.815 \pm 0.015$ | $0.958 \pm 0.003$ | $263.081 \pm 62.775$ | $\mathbf{0.711} \pm 0.146$ | $47.59$ |
| BREPMI | $0.342 \pm 0.013$ | $0.622 \pm 0.026$ | $134.263 \pm 31.441$ | $1.067 \pm 0.208$ | $105.489$ |
| Mirror (black) | $\mathbf{0.611} \pm 0.051$ | $\mathbf{0.862} \pm 0.018$ | $198.609 \pm 40.255$ | $\mathbf{1.049} \pm 0.192$ | $92.413$ |
| C2FMI | $0.414 \pm 0.017$ | $0.686 \pm 0.018$ | $439.659 \pm 93.688$ | $1.592 \pm 0.249$ | $\mathbf{47.317}$ |
| LOKT | $0.328 \pm 0.004$ | $0.553 \pm 0.010$ | $\mathbf{126.964} \pm 36.434$ | $1.122 \pm 0.284$ | $127.709$ |

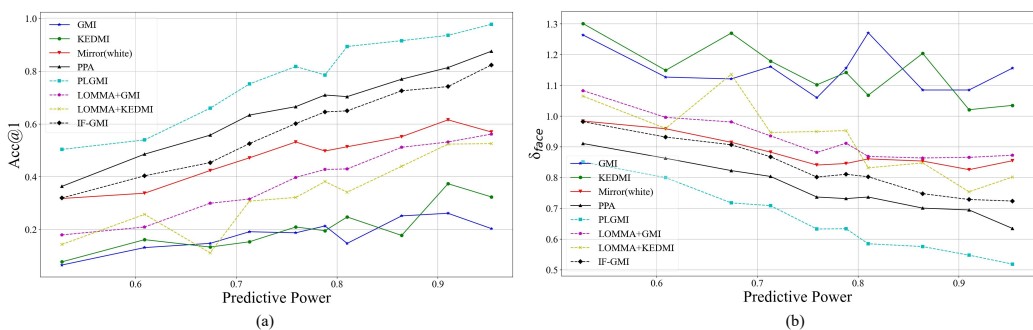

(a)                                                    (b)

Figure 4: Comparison across ResNet-152 with varied predictive power. (a) The incremental trend of $Acc@1$ metric on different attack methods. (b) The decreasing trend of $\delta_{face}$ metric on different attack methods.

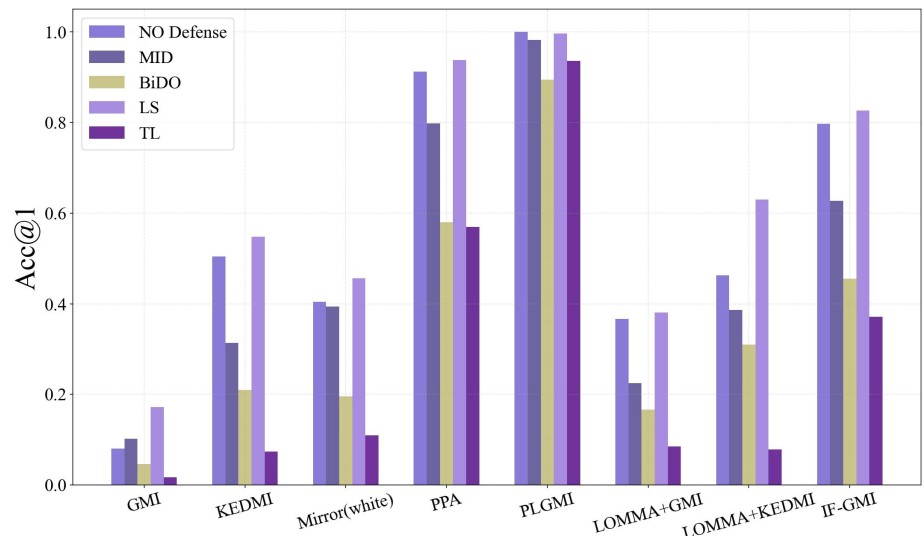

Figure 5: Evaluation on multiple MI defense strategies.

## 4.4 Evaluation on Different Model Predictive Power

The predictive power of the target model is a crucial factor in determining the effectiveness of MI attacks. Previous work (Zhang et al., 2020a) has conducted preliminary experimental validations on simple networks (*e.g.* LeNet (LeCun et al., 1989)), demonstrating that the performance of the first GAN-based attack GMI is influenced by the predictive power of the target model. Therefore, we evaluate the state-of-the-art MI attacks on target models with varied predictive power to further validate the consistency of this characteristic in the recent attacks. The resolution of datasets is set to $64 \times 64$. The evaluation is stated in Fig. 4.

The comparison in Fig. 4 reveals most MI attacks maintain the trend that higher predictive power contributes to better attack performance, which is consistent with the aforementioned characteristic. Specifically, the earlier attack methods (*e.g.*, GMI and KEDMI) presents more fluctuation on the trend across different predictive power, while the recent attack methods (*e.g.*, PLGMI and PPA) show in more stable trend when predictive power increases. This indicates the predictive power of target models plays a crucial role in measuring the performance of MI attacks. Thus one can expect lower privacy leakage in the robust model by balancing the accuracy-privacy trade-off. This study of the predictive power of target models illustrates another useful aspect of our MIBench.

### 4.5 EVALUATION ON DEFENSE STRATEGIES

This analysis concentrates on the robustness of MI attacks when applied to target models with defense strategies. Notably, we select the first 100 classes subset from FaceScrub as the target dataset due to the time cost. The assessment results are listed in Fig.5. The configuration of defenses is set following the official parameters, as detailed in Table 8.

Overall, the TL (Ho et al., 2024) achieves the state-of-the-art decrease in *Accuracy*. However, advanced attack methods have overcome current defense strategies to some extent, such as PLG and PPA. Additionally, some older defense strategies (*e.g.* MID (Wang et al., 2021c)) are no longer effective against the latest attacks. From Fig.5, we observe that LS (Struppek et al., 2023) exhibits unexpectedly poor performance in $Acc@1$ metric while it was recently published in the top-tier conference. The potential reason might be the utilized target classifier with relatively high test accuracy, as validated in the Sec. C.4. Furthermore, we conduct further experiments with PPA as the attack method against ResNet-152 trained on high resolution (224×224) scenario, proving that this phenomenon can be extended into other defenses. The results are listed in the Table 5, demonstrating that all the defenses become invalid even with the recommended parameters. More in-depth evaluation on defenses is exhibited in Sec C.5.

Combining this phenomenon with the above experiment on predictive power, our empirical analysis indicates that the leaked information is strongly correlated to the model prediction accuracy and current defenses cannot effectively reduce the privacy information without sacrifice of model performance. Our findings emphasize that more reliable and stable defense strategies should be studied due to the fact that high model prediction accuracy is crucial for application of AI technology.

Table 5: Extensive evaluation on multiple MI defense strategies.

| Method | Hyperparameters | Test Acc | ↑ **Acc**@1 | ↑ **Acc**@5 | ↓ $\delta_{eval}$ | ↓ $\delta_{face}$ | ↓ **FID** |
|---|---|---|---|---|---|---|---|
| NO Defense | - | 98.510 | 0.972 | 0.990 | 307.714 | 0.588 | 50.259 |
| MID | $\alpha = 0.005$ | 96.760 | 1.000 | 1.000 | 273.687 | 0.517 | 49.239 |
|  | $\alpha = 0.01$ | 95.680 | 0.990 | 1.000 | 276.889 | 0.526 | 51.227 |
| BiDO | $\alpha = 0.01, \beta = 0.1$ | 98.030 | 0.986 | 0.996 | 306.827 | 0.554 | 50.943 |
|  | $\alpha = 0.05, \beta = 0.5$ | 97.430 | 0.968 | 0.996 | 320.191 | 0.594 | 50.132 |
| TL | $\alpha = 0.4$ | 97.620 | 0.994 | 1.000 | 282.394 | 0.513 | 53.023 |
|  | $\alpha = 0.5$ | 97.530 | 0.982 | 0.996 | 306.528 | 0.561 | 51.489 |

## 5 CONCLUSION

In this paper, we develop *MIBench*, a comprehensive, unified, and reliable benchmark, and provide an extensible and reproducible toolbox for researchers. To the best of our knowledge, this is the first benchmark and first open-source toolbox in the MI field. Our benchmark encompasses 16 of the state-of-the-art MI attack methods and defense strategies and more algorithms will be continually updated. Based on the implemented toolbox, we establish a consistent experimental environment and conducted extensive experimental analyses to facilitate fair comparison between different methods. In our experiments, we explore the impact of multiple settings, such as different image resolutions, model predictive power and defense. With in-depth analysis, we have identified new insights and proposed potential solutions to alleviate them.

**Societal Impact and Ethical Considerations.** A potential negative impact of our benchmark could be malicious users leveraging the implemented attack methods to reconstruct private data from public system. To alleviate this potential dilemma, a cautious approach for data users is to adopt robust and reliable defense strategies, as shown in the Sec.4.5 of our paper. Additionally, establishing access permissions and limiting the number of visits for each user is crucial to build responsible AI systems, thereby alleviating the potential contradictions with individual data subjects.

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

APPENDIX TABLE

## A  BENCHMARK DETAILS

### A.1  DETAILS OF DATA PROCESSING

This section introduces three types of dataset processing implemented by our toolbox, including data pre-processing, dataset splitting and dataset synthesis.

#### A.1.1  DATA PRE-PROCESSING.

Data pre-processing aligns face images to ensure consistency across datasets. Users can customize the transformations to be used for data pre-processing. We also provide default processing for four datasets in high and low versions of the resolution. Low resolution versions include $64 \times 64$ and $112 \times 112$, and high resolution versions include $224 \times 224$ and $299 \times 299$. Here are the default pre-processing method for the four datasets.

- **CelebA (Liu et al., 2015).** For the low version, a center-croped with a crop size $108 \times 108$ and a resize function is applied to each sample from the origin images. For the high version, only a direct resize will be used in the images pre-processed from HD-CelebA-Cropper (He, 2020).
- **FaceScrub (Ng & Winkler, 2014).** In the low-resolution version, the original data is center cropped at $54/64$ and then scaled, and the high-resolution is scaled directly.
- **FFHQ (Karras et al., 2019) & MetFaces (Karras et al., 2020a).** In low and high resolution versions, we center cropped the data with $88/128$ and $800/1024$ factors, respectively, and scaled to the specified resolution.

#### A.1.2  DATASET SPLITTING.

This step is used to model training, including classifiers and conditional generators. For labeled datasets like CelebA and FaceScrub, we provide a fixed partition to slice the training and test sets. For unlabeled datasets such as FFHQ and MetFaces, we provide a script for pseudo-labeling all images by computing the scores. The strategy of PLGMI (Yuan et al., 2023a) for the score calculation is contained in the examples of our toolbox.

#### A.1.3  DATASET SYNTHESIS.

Some methods use synthetic datasets for model training. We provide a script to synthesis datasets by pre-trained GANs according to LOKT (Nguyen et al., 2023a).

### A.2  IMPLEMENTATION OF ATTACK MODELS.

We implemented $64 \times 64$ and $256 \times 256$ versions for all the custom attack models defined by the official code of implemented algorithms. For those algorithms that use StyleGAN2 (Karras et al., 2020b), we provide a wrapper for loading the official model and adapting it to our attack process.

### A.3  IMPLEMENTATION OF CLASSIFIER TRAINING & DEFENSE METHODS

Our toolbox supports the training of a wide range of classifiers, including those used by the official code of most implemented algorithms, as well as those supported by TorchVision.

To ensure consistency across different defense strategies, we provide a unified framework for classifier training. Here we present the general idea of implemented methods.

1. **NO Defense**. The classifiers are trained with the cross-entropy loss function.
2. **MID (Wang et al., 2021c).** It applies a Gaussian perturbation to the features in front of the classification head to reduce the mutual information of the model inputs and outputs. According to the official code, the method can also be called *VIB*, and we adopt this name in our toolbox. The hyperparameter $\alpha$ controls the factor of the Gaussian perturbation term.

3. **BiDO (Peng et al., 2022).** It adds the regular term loss (*COCO* or *HSIC*) function so that the intermediate features decrease the mutual information with the input and increase the mutual information with the output. It has two hyperparameters, $\alpha$ and $\beta$. The former is the factor of the regular term loss between inputs and features, and the latter means that between features and model outputs.

4. **Label Smoothing (LS) (Struppek et al., 2023).** It adds an label smoothing term with a negative smoothing factor $\alpha$ into the cross-entropy loss.

5. **Transfer Learning (TL) (Ho et al., 2024).** This method uses a pre-trained model for fine-tuning. The parameters of the previous layers are frozen and those of the later layers will be fine-tuned. In our implementation, the hyperparameter $\alpha$ is defined as a ratio of the number of frozen layers.

In order to fairly compare the various defense methods, all defense methods are trained using the same pre-trained model in our experiments.

### A.4 DETAILS OF ATTACK PROCESS

The attack process follows a sequential workflow, containing **latent vectors sampling**, **labeled latent vectors selection**, **optimization** and **final images selection**. Here are the details of the workflow.

**Latent Vectors Sampling.** This step generate random latent vectors and distributes them for each attack target. Most attack methods use a random distribution strategy. Mirror, PPA and BREP calculate the score of each latent vector corresponding to each label, and for each label the vectors with the highest scores are selected.

**Labeled Latent Vectors Selection (Optional).** The previous step distributes latent vectors for each label, and this step optimizes the latent vectors by calculating the scores of the latent vectors corresponding to the labels and selecting the few vectors with the highest scores. Although currently there are no algorithms use this step, it can be added into the attack algorithms that use conditional generators, e.g. PLGMI (Yuan et al., 2023a) and LOKT (Nguyen et al., 2023a).

**Optimization.** Optimization is the key step for the attack process, it accepts the initialized latent vectors and attack labels as input and outputs the optimized inverted images. We provide several kinds of optimization strategies of each attack method as follows.

1. **Simple White-Box Optimization.** An optimizer for attack algorithms that use the gradient except KEDMI (Chen et al., 2021b) and VMI (Wang et al., 2021a). It optimize and generate an image for each input vector. Some implement details are displayed in Table 6.

2. **Variance White-Box Optimization.** It optimizes results from a Gaussian distribution of latent vectors corresponding to the target labels, and images are generated by random sampling from the optimized distribution. It is used by KEDMI (Chen et al., 2021b).

3. **Miner WhiteBox Optimization.** This optimizer aims to iteratively update parameters of networks that are utilized to produce high-quality latent vectors, such as Flow models (Xu et al., 2022). It is used by VMI (Wang et al., 2021a).

4. **Genetic Optimization.** An optimizer using genetic algorithms for optimization. The black-box version of Mirror (An et al., 2022) and C2FMI (Ye et al., 2023) use this optimization strategy.

5. **BREP Optimization.** A specific optimizer for BREPMI algorithm (Kahla et al., 2022). It uses a boundary repelling strategy for gradient simulation.

6. **Reinforcement Learning Optimization.** Optimizing the latent vectors via reinforcement learning. Used by RLB attack method (Han et al., 2023).

**Final Images Selection (Optional).** This step works by calculating the scores of each image and selecting the part of the image with the highest score as the result of the attack. It is used by PPA (Struppek et al., 2022).

Table 6: Overview of implement of different attack methods that use White-Box Optimization.

| Method | Latent Optimizer | Identity Loss | Prior Loss | Image Augment |
|--------|------------------|---------------|------------|---------------|
| GMI | Momentum SGD | CE | Discriminator | ✗ |
| KEDMI | Adam | CE | Discriminator | ✗ |
| Mirror | Adam | CE | - | ✔ |
| PPA | Adam | Poincaré | - | ✔ |
| PLGMI | Adam | Max Margin | - | ✔ |
| LOMMA | Adam | Logit | Feature Distance | ✗ |
| IF-GMI | Adam | Poincaré | - | ✔ |
| LOKT | Adam | Max Margin | - | ✔ |

## A.5 DETAILS OF EVALUATION

We provide the following four evaluation metrics to evaluate the effectiveness of the attack.

1. **Classification Accuracy.** The metric uses a given classifier to classify the inverted images and measures the top-1 and top-5 accuracy for target labels. The higher the reconstructed samples achieve attack accuracy on another classifier trained with the same dataset, the more private information in the dataset can be considered to be exposed (Zhang et al., 2020a).

2. **Feature Distance.** The feature is defined as the output of the given classifier's penultimate layer. We compute the shortest feature $l_2$ distance between inverted images and private samples for each class and calculate the average distance. Smaller feature distance means more similar features to the private image.

3. **Fréchet Inception Distance (FID).** FID (Heusel et al., 2017) is commonly used to evaluate the generation quality of GANs. The lower FID score shows higher inter-class diversity and realism (Wang et al., 2021a).

4. **Sample Diversity.** The metric contains Precision-Recall (Kynkäänniemi et al., 2019) and Density-Coverage (Naeem et al., 2020) scores. Higher values indicate greater intra-class diversity of the inverted images.

## B EXPERIMENTAL DETAILS

This section describes the setup and details of the experiments in this paper.

### B.1 EXPERIMENT SETTINGS

We conducted experiments at both low and high resolution scenarios. For the low resolution experiments, we employed classifiers with a resolution of $64 \times 64$ as the target models, and an ResNet-50 with a resolution of $112 \times 112$ served as the evaluation model. In the high resolution experiments, we used classifiers with a resolution of $224 \times 224$ as the target models, with an Inception-v3 model having a resolution of $299 \times 299$ as the evaluation model. Additionally, the computation resources utilized in our experiments including $16\times$ NVIDIA RTX 4090 and $8\times$ NVIDIA A100.

### B.2 DATASETS

The datasets used in our experiments are categorized into two types: public datasets and private datasets. The private datasets are used to train the target and evaluation models. Specifically, we selected 1000 identities with the most images from the CelebA dataset and all 530 identities from the FaceScrub dataset as our private datasets.

The public datasets serve as a priori knowledge for the attacker to train the generator or to extract features of real faces. For the low-resolution experiments, we used FFHQ and the images from the CelebA dataset that are not included in the private dataset as our public datasets. For the high-resolution experiments, FFHQ and MetFaces were chosen as the public datasets. Note that MetFaces is an image dataset of 1336 human faces extracted from the Metropolitan Museum of Art Collection. It has a huge distribution shift with real human faces, which makes model inversion attack algorithms encounter great challenges.

The preprocessing of these datasets is described in A.1.1.

## B.3 CLASSIFIERS

For the attack experiments, we trained multiple classifiers as target models and evaluation models, as detailed in Table 7. For the defense experiments, we trained the ResNet-152 model on the FaceScrub dataset using various defense methods, as outlined in Table 8.

Table 7: Overview of target and evaluation models used in attack experiments.

| Dataset | Model | Resolution | Test Acc |
|---|---|---|---|
| CelebA | VGG-16 | $64 \times 64$ | 0.8826 |
| | ResNet-152 | $64 \times 64$ | 0.9371 |
| | ResNet-50 | $112 \times 112$ | 0.9588 |
| | ResNet-152 | $224 \times 224$ | 0.9003 |
| | Inception-v3 | $299 \times 299$ | 0.9216 |
| FaceScrub | VGG-16 | $64 \times 64$ | 0.8785 |
| | ResNet-152 | $64 \times 64$ | 0.9825 |
| | ResNet-50 | $112 \times 112$ | 0.9938 |
| | ResNet-152 | $224 \times 224$ | 0.9225 |
| | ResNeSt-101 | $224 \times 224$ | 0.9329 |
| | Inception-v3 | $299 \times 299$ | 0.9445 |

Table 8: Overview of IR-152 trained with FaceScrub dataset in different defense methods. The definition of hyperparameters are described in A.3.

| Defense Method | Hyperparameters | Test Acc |
|---|---|---|
| No Defense | - | 0.9825 |
| MID | $\alpha = 0.01$ | 0.9824 |
| BiDO | $\alpha = 0.01, \beta = 0.1$ | 0.9401 |
| LS | $\alpha = -0.05$ | 0.9802 |
| TL | $\alpha = 0.5$ | 0.9536 |

## B.4 EVALUATION.

The definitions of the evaluation metrics are detailed in Section A.5. Here, we present the specific details of the metrics used in the experiments.

For the Classification Accuracy and Feature Distance metrics, we evaluate the attack results using another classifier pre-trained on the same dataset as the target model: ResNet-50 for low-resolution settings and Inception-v3 for high-resolution settings, denoted as **Acc** and $\delta_{eval}$. Additionally, an Inception-v1 model pre-trained on a large face dataset, VGGFace2, is used to calculate the feature distance, measuring the realism of the inverted images, denoted as $\delta_{face}$.

For FID, Precision-Recall, and Density-Coverage scores, we follow the experimental setup of existing papers. We use Inception-v3, pre-trained on ImageNet, to extract the features of images and participate in the score calculation.

## C MORE EXPERIMENTAL RESULTS

## C.1 SAMPLE DIVERSITY

Following the settings of Section 4.2 and 4.3, we computed the Precision-Recall (Kynkäänniemi et al., 2019) and Density-Coverage (Kynkäänniemi et al., 2019) to evaluate the intra-class diversity

for each attack method. The results are presented in Table 9, 10, 11 and 12. It tends to be that the attacks with stronger GAN priors get higher scores, such as Mirror, C2FMI and PPA.

Table 9: Comparison between white-box MI attacks on low resolution scenario.

| Method | ↑ Precision | ↑ Recall | ↑ Density | ↑ Coverage |
|---|---|---|---|---|
| GMI | $0.025 \pm 0.079$ | $0.797 \pm 0.200$ | $0.008 \pm 0.018$ | $0.019 \pm 0.037$ |
| KEDMI | $0.065 \pm 0.159$ | $0.055 \pm 0.158$ | $0.017 \pm 0.051$ | $0.025 \pm 0.062$ |
| Mirror(white) | $\mathbf{0.205} \pm 0.232$ | $0.444 \pm 0.304$ | $\mathbf{0.067} \pm 0.095$ | $\mathbf{0.133} \pm 0.148$ |
| PPA | $0.149 \pm 0.199$ | $0.499 \pm 0.273$ | $0.043 \pm 0.073$ | $0.089 \pm 0.122$ |
| PLGMI | $0.048 \pm 0.116$ | $0.302 \pm 0.283$ | $0.014 \pm 0.038$ | $0.030 \pm 0.064$ |
| LOMMA+GMI | $0.062 \pm 0.123$ | $\mathbf{0.828} \pm 0.187$ | $0.018 \pm 0.043$ | $0.040 \pm 0.080$ |
| LOMMA+KEDMI | $0.061 \pm 0.187$ | $0.000 \pm 0.008$ | $0.019 \pm 0.052$ | $0.023 \pm 0.052$ |
| IF-GMI | $0.129 \pm 0.191$ | $0.585 \pm 0.279$ | $0.039 \pm 0.068$ | $0.081 \pm 0.115$ |

Table 10: Comparison between black-box MI attacks on low resolution scenario.

| Method | ↑ Precision | ↑ Recall | ↑ Density | ↑ Coverage |
|---|---|---|---|---|
| BREP | $0.048 \pm 0.131$ | $0.249 \pm 0.309$ | $0.013 \pm 0.030$ | $0.023 \pm 0.051$ |
| Mirror(black) | $0.085 \pm 0.147$ | $\mathbf{0.489} \pm 0.294$ | $\mathbf{0.262} \pm 0.043$ | $\mathbf{0.059} \pm 0.083$ |
| C2F | $\mathbf{0.118} \pm 0.234$ | $0.029 \pm 0.125$ | $0.037 \pm 0.078$ | $0.053 \pm 0.089$ |
| LOKT | $0.051 \pm 0.129$ | $0.232 \pm 0.292$ | $0.013 \pm 0.032$ | $0.027 \pm 0.063$ |

Table 11: Comparison between white-box MI attacks on high resolution scenario.

| Method | ↑ Precision | ↑ Recall | ↑ Density | ↑ Coverage |
|---|---|---|---|---|
| GMI | $0.033 \pm 0.086$ | $\mathbf{0.758} \pm 0.248$ | $0.005 \pm 0.011$ | $0.013 \pm 0.028$ |
| KEDMI | $0.029 \pm 0.116$ | $0.055 \pm 0.170$ | $0.006 \pm 0.016$ | $0.010 \pm 0.027$ |
| Mirror(white) | $0.217 \pm 0.236$ | $0.350 \pm 0.292$ | $0.048 \pm 0.072$ | $0.092 \pm 0.099$ |
| PPA | $\mathbf{0.259} \pm 0.243$ | $0.322 \pm 0.266$ | $\mathbf{0.060} \pm 0.075$ | $\mathbf{0.112} \pm 0.102$ |
| PLGMI | $0.019 \pm 0.099$ | $0.002 \pm 0.025$ | $0.005 \pm 0.015$ | $0.007 \pm 0.018$ |
| LOMMA+GMI | $0.023 \pm 0.074$ | $0.514 \pm 0.333$ | $0.006 \pm 0.013$ | $0.013 \pm 0.028$ |
| LOMMA+KEDMI | $0.033 \pm 0.130$ | $0.003 \pm 0.041$ | $0.008 \pm 0.026$ | $0.011 \pm 0.024$ |
| IF-GMI | $0.154 \pm 0.200$ | $0.339 \pm 0.287$ | $0.035 \pm 0.052$ | $0.068 \pm 0.074$ |

Table 12: Comparison between black-box MI attacks on high resolution scenario.

| Method | ↑ Precision | ↑ Recall | ↑ Density | ↑ Coverage |
|---|---|---|---|---|
| BREP | $0.041 \pm 0.121$ | $\mathbf{0.160} \pm 0.286$ | $0.008 \pm 0.024$ | $0.014 \pm 0.029$ |
| Mirror(black) | $0.011 \pm 0.055$ | $0.115 \pm 0.201$ | $0.004 \pm 0.010$ | $0.008 \pm 0.024$ |
| C2F | $\mathbf{0.119} \pm 0.227$ | $0.026 \pm 0.115$ | $\mathbf{0.024} \pm 0.048$ | $\mathbf{0.036} \pm 0.063$ |
| LOKT | $0.014 \pm 0.083$ | $0.023 \pm 0.125$ | $0.004 \pm 0.013$ | $0.006 \pm 0.016$ |

## C.2 EVALUATION ON DIFFERENT TARGET CLASSIFIERS

In addition to attacking IR-152 and ResNet-152 in Section 4.2 and 4.3, we extend our experiments on more different target classifiers. We evaluate attacks on VGG-16 (Simonyan & Zisserman, 2014) and ResNeSt-101 (Zhang et al., 2022a) in low and high resolution settings, respectively. The results are presented in Table 13 and 14.

Besides the aforementioned CNN-based classifiers, we further analyze MI attacks on ViT-based models. Table 15 presents further experiments on the MaxViT (Tu et al., 2022) under the high

Table 13: Comparison between white-box MI attacks against VGG-16 on low resolution scenario.

| Method | $\uparrow$ **Acc@1** | $\uparrow$ **Acc@5** | $\downarrow \delta_{eval}$ | $\downarrow \delta_{face}$ | $\downarrow$ **FID** |
|---|---|---|---|---|---|
| GMI | $0.013 \pm 0.003$ | $0.046 \pm 0.018$ | $2565.303 \pm 290.350$ | $1.352 \pm 0.142$ | 62.205 |
| KEDMI | $0.074 \pm 0.008$ | $0.190 \pm 0.013$ | $2553.729 \pm 412.648$ | $1.147 \pm 0.254$ | 91.953 |
| Mirror(white) | $0.061 \pm 0.007$ | $0.165 \pm 0.006$ | $2358.875 \pm 347.703$ | $1.111 \pm 0.174$ | 37.605 |
| PPA | $0.263 \pm 0.019$ | $0.461 \pm 0.023$ | $2018.148 \pm 377.491$ | $0.874 \pm 0.160$ | **33.226** |
| PLGMI | $\mathbf{0.465} \pm 0.019$ | $\mathbf{0.683} \pm 0.008$ | $\mathbf{1914.942} \pm 409.569$ | $\mathbf{0.762} \pm 0.174$ | 81.093 |
| LOMMA+GMI | $0.091 \pm 0.026$ | $0.216 \pm 0.047$ | $2503.465 \pm 288.728$ | $1.060 \pm 0.153$ | 60.650 |
| LOMMA+KEDMI | $0.233 \pm 0.009$ | $0.418 \pm 0.011$ | $2258.070 \pm 480.906$ | $0.912 \pm 0.205$ | 66.410 |
| IF-GMI | $0.208 \pm 0.010$ | $0.391 \pm 0.021$ | $2102.656 \pm 369.571$ | $0.928 \pm 0.172$ | 35.816 |

Table 14: Comparison between white-box MI attacks against ResNeSt-101 on high resolution scenario.

| Method | $\uparrow$ **Acc@1** | $\uparrow$ **Acc@5** | $\downarrow \delta_{eval}$ | $\downarrow \delta_{face}$ | $\downarrow$ **FID** |
|---|---|---|---|---|---|
| GMI | $0.069 \pm 0.011$ | $0.191 \pm 0.036$ | $135.290 \pm 22.961$ | $1.339 \pm 0.135$ | 124.880 |
| KEDMI | $0.153 \pm 0.013$ | $0.353 \pm 0.012$ | $143.155 \pm 32.520$ | $1.258 \pm 0.245$ | 140.533 |
| Mirror(white) | $0.380 \pm 0.027$ | $0.684 \pm 0.021$ | $193.275 \pm 29.316$ | $1.032 \pm 0.161$ | 58.437 |
| PPA | $0.904 \pm 0.008$ | $0.984 \pm 0.002$ | $159.986 \pm 27.495$ | $0.781 \pm 0.157$ | **44.966** |
| PLGMI | $\mathbf{0.931} \pm 0.006$ | $\mathbf{0.988} \pm 0.003$ | $147.914 \pm 40.333$ | $\mathbf{0.753} \pm 0.177$ | 92.755 |
| LOMMA+GMI | $0.577 \pm 0.134$ | $0.770 \pm 0.123$ | $\mathbf{131.040} \pm 28.470$ | $1.042 \pm 0.165$ | 133.604 |
| LOMMA+KEDMI | $0.373 \pm 0.008$ | $0.615 \pm 0.007$ | $148.923 \pm 42.489$ | $1.129 \pm 0.285$ | 139.433 |
| IF-GMI | $0.736 \pm 0.013$ | $0.920 \pm 0.011$ | $236.910 \pm 49.451$ | $0.647 \pm 0.133$ | 45.759 |

Table 15: Comparison between white-box MI attacks against MaxViT on high resolution scenario.

| Method | $\uparrow$ **Acc@1** | $\uparrow$ **Acc@5** | $\downarrow \delta_{eval}$ | $\downarrow \delta_{face}$ | $\downarrow$ **FID** |
|---|---|---|---|---|---|
| GMI | $0.018 \pm 0.007$ | $0.080 \pm 0.021$ | $260.084 \pm 71.029$ | $1.406 \pm 0.131$ | 154.447 |
| KEDMI | $0.112 \pm 0.007$ | $0.270 \pm 0.028$ | $261.827 \pm 73.975$ | $1.117 \pm 0.224$ | 148.083 |
| Mirror | $0.146 \pm 0.034$ | $0.346 \pm 0.066$ | $286.339 \pm 47.047$ | $1.034 \pm 0.158$ | 80.136 |
| PPA | $0.522 \pm 0.020$ | $0.758 \pm 0.010$ | $237.410 \pm 41.175$ | $0.776 \pm 0.118$ | 66.023 |
| PLGMI | $0.322 \pm 0.035$ | $0.574 \pm 0.042$ | $261.860 \pm 55.469$ | $0.772 \pm 0.137$ | 153.054 |
| LOMMA+GMI | $0.374 \pm 0.077$ | $0.620 \pm 0.058$ | $244.566 \pm 48.069$ | $0.920 \pm 0.148$ | 138.875 |
| LOMMA+KEDMI | $0.294 \pm 0.014$ | $0.552 \pm 0.022$ | $260.002 \pm 71.687$ | $0.910 \pm 0.217$ | 150.214 |
| IF-GMI | $0.408 \pm 0.012$ | $0.669 \pm 0.020$ | $230.251 \pm 42.734$ | $0.801 \pm 0.139$ | 45.625 |

resolution (224×224) scenario. Other experimental settings are continuous with the main paper, with FFHQ as the public dataset and FaceScrub as the private dataset. The test accuracy of the target MaxViT is 94.61%.

## C.3 EVALUATION ON MORE COMBINATION OF DATASETS

Evaluations in the Section 4 are conducted under the same dataset combination of FFHQ as the public dataset and FaceScrub as the private dataset. Therefore, we design more combination of datasets in this part to further assess the transferability of different attacks. The results are listed in Table 16 and 17. The visual results are shown in Figure 6 and 7.

Except for the typical face classification task, we have conducted more experiments on the dog breed classification task under the high resolution (224×224) scenario, which includes two non-facial datasets, Stanford Dogs (Dataset, 2011) and Animal Faces-HQ Dog (AFHQ) (Choi et al., 2020). The public dataset is AFHQ while the private dataset is Stanford Dogs. The target model is ResNet-152 of 77.45% test accuracy, which follows the same setting in PPA. Table 18 lists the evaluation results.

Table 16: Comparison between white-box MI attacks with FFHQ prior against ResNet-152 pre-trained on CelebA on high resolution scenario.

| Method | ↑ **Acc@1** | ↑ **Acc@5** | ↓ $\delta_{eval}$ | ↓ $\delta_{face}$ | ↓ **FID** |
|---|---|---|---|---|---|
| GMI | $0.050 \pm 0.008$ | $0.171 \pm 0.031$ | $216.614 \pm 36.221$ | $1.248 \pm 0.138$ | 108.217 |
| KEDMI | $0.174 \pm 0.013$ | $0.391 \pm 0.011$ | $247.112 \pm 55.473$ | $1.113 \pm 0.221$ | 119.760 |
| Mirror(white) | $0.367 \pm 0.026$ | $0.661 \pm 0.024$ | $286.668 \pm 47.261$ | $0.973 \pm 0.166$ | 63.261 |
| PPA | $0.936 \pm 0.008$ | $0.987 \pm 0.002$ | $233.474 \pm 50.366$ | $\mathbf{0.711} \pm 0.148$ | **46.339** |
| PLGMI | $\mathbf{0.953} \pm 0.007$ | $\mathbf{0.992} \pm 0.002$ | $261.210 \pm 58.636$ | $0.726 \pm 0.167$ | 151.119 |
| LOMMA+GMI | $0.664 \pm 0.121$ | $0.815 \pm 0.100$ | $\mathbf{207.854} \pm 39.254$ | $0.938 \pm 0.162$ | 109.383 |
| LOMMA+KEDMI | $0.222 \pm 0.005$ | $0.411 \pm 0.007$ | $229.407 \pm 65.371$ | $1.178 \pm 0.346$ | 145.272 |
| IF-GMI | $0.986 \pm 0.003$ | $0.999 \pm 0.001$ | $222.919 \pm 52.073$ | $0.614 \pm 0.140$ | 37.408 |

Table 17: Comparison between white-box MI attacks with Metfaces prior against ResNet-152 pre-trained on CelebA on high resolution scenario.

| Method | ↑ **Acc@1** | ↑ **Acc@5** | ↓ $\delta_{eval}$ | ↓ $\delta_{face}$ | ↓ **FID** |
|---|---|---|---|---|---|
| GMI | $0.008 \pm 0.003$ | $0.046 \pm 0.008$ | $\mathbf{209.264} \pm 45.093$ | $1.392 \pm 0.149$ | 217.151 |
| KEDMI | $0.002 \pm 0.001$ | $0.011 \pm 0.002$ | $250.805 \pm 62.654$ | $1.561 \pm 0.232$ | 276.504 |
| Mirror(white) | $0.100 \pm 0.007$ | $0.265 \pm 0.009$ | $357.719 \pm 52.080$ | $1.261 \pm 0.194$ | 78.541 |
| PPA | $\mathbf{0.463} \pm 0.020$ | $\mathbf{0.726} \pm 0.020$ | $305.953 \pm 57.145$ | $\mathbf{1.074} \pm 0.203$ | **72.372** |
| PLGMI | $0.126 \pm 0.003$ | $0.274 \pm 0.005$ | $220.139 \pm 41.739$ | $1.126 \pm 0.218$ | 393.518 |
| LOMMA+GMI | $0.061 \pm 0.019$ | $0.140 \pm 0.032$ | $214.122 \pm 54.770$ | $1.370 \pm 0.228$ | 245.013 |
| LOMMA+KEDMI | $0.006 \pm 0.001$ | $0.013 \pm 0.001$ | $245.896 \pm 63.101$ | $1.630 \pm 0.253$ | 320.662 |
| IF-GMI | $0.934 \pm 0.010$ | $0.988 \pm 0.003$ | $235.986 \pm 46.216$ | $0.768 \pm 0.162$ | 73.375 |

Table 18: Comparison between white-box MI attacks with AFHQ prior against ResNet-152 pre-trained on Stanford Dogs on high resolution scenario.

| Method | ↑ **Acc@1** | ↑ **Acc@5** | ↓ $\delta_{eval}$ | ↓ **FID** |
|---|---|---|---|---|
| GMI | $0.068 \pm 0.031$ | $0.226 \pm 0.026$ | $88.447 \pm 15.990$ | 210.543 |
| KEDMI | $0.606 \pm 0.027$ | $0.830 \pm 0.032$ | $66.521 \pm 16.994$ | 134.513 |
| Mirror | $0.656 \pm 0.058$ | $0.848 \pm 0.017$ | $142.580 \pm 49.569$ | 77.485 |
| PPA | $0.906 \pm 0.026$ | $0.990 \pm 0.006$ | $121.571 \pm 45.929$ | 58.479 |
| PLGMI | $0.216 \pm 0.016$ | $0.504 \pm 0.022$ | $86.629 \pm 20.109$ | 238.115 |
| LOMMA+GMI | $0.302 \pm 0.103$ | $0.486 \pm 0.126$ | $84.761 \pm 24.458$ | 198.523 |
| LOMMA+KEDMI | $0.838 \pm 0.007$ | $0.968 \pm 0.007$ | $58.225 \pm 22.527$ | 97.301 |
| IF-GMI | $0.947 \pm 0.008$ | $0.993 \pm 0.003$ | $147.845 \pm 66.393$ | 48.972 |

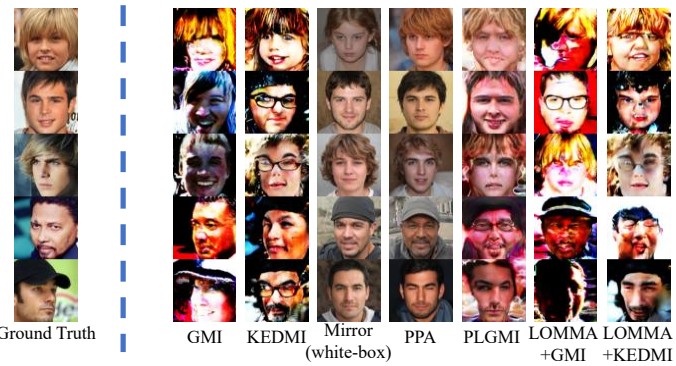

Ground Truth | GMI | KEDMI | Mirror (white-box) | PPA | PLGMI | LOMMA +GMI | LOMMA +KEDMI

Figure 6: Visual comparison between different MI attacks with FFHQ prior.

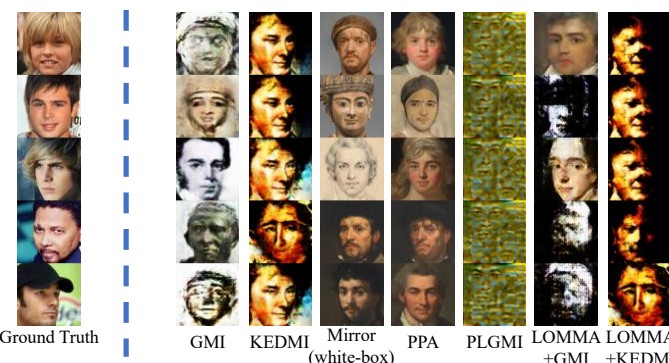

Ground Truth     GMI   KEDMI   Mirror (white-box)   PPA   PLGMI   LOMMA +GMI   LOMMA +KEDMI

Figure 7: Visual comparison between different MI attacks with MetFaces prior.

## C.4 EVALUATION ON LS DEFENSE METHOD

To evaluate the LS defense method, we follow the settings in the official paper (Struppek et al., 2023) and evaluate it in more settings, shown in Table 19. To better evaluate the effectiveness of LS defense algorithms by making the undefended classifiers slightly less accurate than the defense-imposed classifiers through an early-stop strategy when training.

In setting A, the public and private datasets are different part of CelebA dataset in low resolution scenario with the GMI as the attack method. In setting B, the public dataset is FFHQ and the private dataset is FaceScrub with the PPA as the attack method. In these two settings, the predictive power is much lower than that describes in Section 4.5. In this case, the LS defense is very effective, making the success rate of the attack drop dramatically.

Table 19: Evaluation on LS defense method on different settings.

| Setting | Defense | Test Acc | $\uparrow$ Acc@1 | $\uparrow$ Acc@5 | $\downarrow \delta_{eval}$ | $\downarrow \delta_{face}$ |
|---|---|---|---|---|---|---|
| A | - | 0.832 | $0.068 \pm 0.054$ | $0.180 \pm 0.126$ | $1949.072 \pm 184.779$ | $1.281 \pm 0.129$ |
| | LS | 0.851 | $0.005 \pm 0.003$ | $0.021 \pm 0.014$ | $1984.984 \pm 244.580$ | $1.472 \pm 0.199$ |
| B | - | 0.861 | $\mathbf{0.826} \pm 0.032$ | $\mathbf{0.965} \pm 0.008$ | $\mathbf{176.483} \pm 33.399$ | $\mathbf{0.844} \pm 0.154$ |
| | LS | **0.869** | $0.320 \pm 0.062$ | $0.602 \pm 0.068$ | $233.413 \pm 43.395$ | $1.107 \pm 0.186$ |

## C.5 VALIDATION FOR DEFENSE METHODS ON MODELS WITH LOW ACCURACY

The results for models with low accuracy are listed in Table 20. Obviously, the defense is valid when the target model has relatively low prediction accuracy.

Table 20: Evaluation on defense methods for models with low accuracy.

| Method | Hyperparameters | Test Acc | $\uparrow$ Acc@1 | $\uparrow$ Acc@5 | $\downarrow \delta_{eval}$ | $\downarrow \delta_{face}$ | $\downarrow$ **FID** |
|---|---|---|---|---|---|---|---|
| NO Defense | - | 92.170 | 0.686 | 0.914 | 262.471 | 0.767 | 68.454 |
| MID | $\alpha = 0.005$ | 88.240 | 0.568 | 0.820 | 246.021 | 0.757 | 69.663 |
| BiDO | $\alpha = 0.01, \beta = 0.1$ | 88.620 | 0.582 | 0.874 | 275.453 | 0.793 | 68.248 |
| TL | $\alpha = 0.4$ | 89.160 | 0.316 | 0.616 | 279.439 | 0.897 | 63.241 |

## C.6 EVALUATION ON DIFFERENT LOSS FUNCTIONS

In recent years, various attack algorithms have attempted to mitigate the effects of gradient vanishing by employing different loss functions. In this part, we investigate the impact of identity loss functions on the success rate of model inversion attacks. Specifically, we adopt PPA with FFHQ prior to attack a ResNet-152 classifier pre-trained on FaceScrub. The comparison of the results is presented

in Table 21. Our findings indicate that the Poincare loss function yields the highest performance without model augmentation, whereas the Logit loss function achieves the best results with model augmentation.

Table 21: Comparision of different identity loss. "+" denotes that the target model is used as the teacher model, and three students models are distilled using the public dataset and jointly involved in the loss calculation. It is called model augmentation in the paper of LOMMA Nguyen et al. (2023b). Logit loss here is implemented via pytorch's NLLLoss.

| Loss Function | $\uparrow$ **Acc@1** | $\uparrow$ **Acc@5** | $\downarrow \delta_{eval}$ | $\downarrow \delta_{face}$ | $\downarrow$ **FID** |
|---|---|---|---|---|---|
| CE | $0.769 \pm 0.032$ | $0.942 \pm 0.012$ | $172.305 \pm 26.753$ | $0.901 \pm 0.140$ | 53.880 |
| Poincaré | $0.913 \pm 0.022$ | $0.986 \pm 0.004$ | $167.532 \pm 28.944$ | $0.774 \pm 0.143$ | 46.246 |
| Max Margin | $0.812 \pm 0.020$ | $0.951 \pm 0.008$ | $169.730 \pm 27.705$ | $0.871 \pm 0.150$ | 51.146 |
| Logit | $0.886 \pm 0.023$ | $0.978 \pm 0.013$ | $170.867 \pm 31.415$ | $0.806 \pm 0.148$ | 45.731 |
| CE$^+$ | $\mathbf{0.946} \pm 0.011$ | $0.992 \pm 0.002$ | $165.461 \pm 29.055$ | $0.785 \pm 0.147$ | 48.564 |
| Poincaré$^+$ | $0.901 \pm 0.006$ | $0.984 \pm 0.004$ | $\mathbf{153.921} \pm 24.542$ | $0.812 \pm 0.158$ | 45.114 |
| Max Margin$^+$ | $0.918 \pm 0.017$ | $0.985 \pm 0.003$ | $165.420 \pm 27.607$ | $0.815 \pm 0.148$ | 49.292 |
| Logit$^+$ | $0.945 \pm 0.007$ | $\mathbf{0.993} \pm 0.003$ | $177.745 \pm 34.030$ | $\mathbf{0.764} \pm 0.159$ | $\mathbf{44.166}$ |

## C.7 More evaluation on VMI, RLBMI and PPA

Considering the high computational overhead of RLBMI and VMI, we only experiment at a low resolution settings for 100 classes. The public and private datasets are different part of CelebA dataset. The results are shown in Table 22.

Table 22: Experimental results of VMI and RLBMI.

| Method | $\uparrow$ **Acc@1** | $\uparrow$ **Acc@5** | $\downarrow \delta_{eval}$ | $\downarrow \delta_{face}$ |
|---|---|---|---|---|
| VMI | $0.168 \pm 0.018$ | $0.273 \pm 0.019$ | $1822.122 \pm 398.948$ | $1.260 \pm 0.397$ |
| RLBMI | $0.780 \pm 0.040$ | $0.920 \pm 0.075$ | $1173.134 \pm 250.088$ | $0.699 \pm 0.049$ |

We also explored the effect of PPA for different number of latent vectors for optimization and the number of iterations. It is presented in Table 23. Note that PPA select top-5 optimized latent vectors as attack results.

Table 23: Experiment result of PPA for different number of latent vectors to optimize and iterations.

| Number of latents | Iterations | $\uparrow$ **Acc@1** | $\uparrow$ **Acc@5** | $\downarrow \delta_{eval}$ | $\downarrow \delta_{face}$ | $\downarrow$ **FID** |
|---|---|---|---|---|---|---|
| 20 | 50 | $0.433 \pm 0.072$ | $0.651 \pm 0.076$ | $1888.342 \pm 478.644$ | $0.898 \pm 0.239$ | 40.138 |
| 50 | 50 | $0.496 \pm 0.067$ | $0.698 \pm 0.048$ | $1804.830 \pm 471.815$ | $0.847 \pm 0.224$ | $\mathbf{38.765}$ |
| 20 | 600 | $\mathbf{0.844} \pm 0.042$ | $\mathbf{0.924} \pm 0.026$ | $\mathbf{1391.261} \pm 396.732$ | $\mathbf{0.658} \pm 0.194$ | 46.246 |

## D Limitations and Future Plans

Our benchmark mainly focuses on GAN-based MI attacks and MI defenses applied in the classifier training stage. We will extend our benchmark to wider range of MI methods, including learning-based MI attacks and MI defenses applied to the classifier output. Furthermore, the concept of MI also pervades across modalities (*e.g.* text (Parikh et al., 2022; Zhang et al., 2023; Carlini et al., 2019; 2021; Yu et al., 2023; Nasr et al., 2023) and graph learning (Zhang et al., 2021; 2022b; Zhou et al., 2023; Wu et al., 2022; He et al., 2021)) beyond computer vision domain, where our benchmark concentrates in current stage. Expansion to new modalities is a promising direction for our benchmark to further explore the privacy threats in other fields, leading to more generalization in the AI security. In addition to developing new algorithms, it is also essential to conduct further research on the MI attacks and defenses to make in-depth analysis about their characteristics and bring valuable new insights.

