# OpenReview forum: "MIBench: A Comprehensive Benchmark for Model Inversion Attack and Defense"
_ICLR.cc/2025/Conference — ICLR 2025 Conference Withdrawn Submission_

### Official Review · Reviewer_sRmV · 2024-10-19

**Soundness:** 4
**Presentation:** 3
**Contribution:** 2
**Rating:** 6
**Confidence:** 5

**Summary:**

This paper presents a unified toolbox for Model Inversion (MI) research, implemented using a modular approach. It incorporates most state-of-the-art MI attacks and defences. Using this toolbox, the authors conduct several studies exploring the effectiveness of various MI attacks and defences, the impact of image resolution, and the correlation between MI performance and model utility.

**Strengths:**

- Overall, I appreciate the paper's contribution on creating a unified MI toolbox, which is very helpful for researchers in MI community.

- Additionally, comprehensive experiments have been conducted, and several results/analysis are provided.

**Weaknesses:**

- From the research perspective, no new concepts have been introduced.

- Despite there are some analysis on MI, the research scopes and results are somewhat trivial or have been discussed in the previous works. For example, the strong correlation of MI performance and model utility is shown in GMI paper [1]; the advantage of MI attacks that employ stronger GAN (i.e., StyleGAN) is discussed in PPA paper [2].

- There is a new defence method [3] that recently published at ECCV-2024 [3]. Of course, I do not expect the authors to include this defence for this submission. However, as a toolbox for MI research, it would be better to discuss it in the related work section and incorporates this defence to the toolbox in the future.

[1] Zhang, Yuheng, et al. "The secret revealer: Generative model-inversion attacks against deep neural networks." CVPR 2020.

[2] Struppek, Lukas, et al. "Plug & play attacks: Towards robust and flexible model inversion attacks." ICML 2022.

[3] Koh, Jun Hao, et al. "On the Vulnerability of Skip Connections to Model Inversion Attacks." ECCV 2024.

**Questions:**

- In Tab. 5, I assume that the authors train the target classifiers to achieve comparable model utilities across MI defences for a fair comparison. In the original MI defence papers, the model utilities to achieve the best performance for each defence are different. I am curious about the results of Tab. 5 when analysing on different model utilities (i.e., from 90-91% instead of 96-98%).  I suggest authors to conduct an ablation study showing how the effectiveness of different defences changes across a range of model utilities

- In Tab. 5, from my understanding, the conclusions are drawn from the first 100 classes of Facescrub due to the time cost. However, this raise a concern if the results are biased to this selected classes. I suggest the authors to conduct this comparison on different set of 100 classes of Facescrub, or on different dataset to further confirm the conclusions.

- The results of PLG-MI in Tab. 4 and Fig. 3 are somewhat surprising. From my understanding, PLG-MI does not leverage powerful GANs like Mirror, PPA, or IF-GMI, yet it achieves the highest MI attack accuracy along with impressive visual outcomes. Could the authors provide further explanation for this?

- The results of IF-GMI in Tab. 4  is also very different from the reported values in the original paper. I believe that it is due to the difference in experimental setup. Could the authors point out this difference?

---

### Official Review · Reviewer_69wB · 2024-10-28

**Soundness:** 2
**Presentation:** 2
**Contribution:** 2
**Rating:** 3
**Confidence:** 4

**Summary:**

Authors propose a unified package for model inversion evaluations that include model training, attacks, defenses, and evaluations. This benchmark focuses on face-recognition related models under both white-box and black-box adversaries, covering attacks that utilize GAN-based techniques. The benchmark also contains multiple datasets and defenses based on training-time modifications.

**Strengths:**

- The paper aims to unify benchmarking by combining multiple attacks and defenses in a central library.

- Based on inspecting the codebase, it seems fairly straightforward to add new attacks/datasets/models, which is crucial as an enable for research in the corresponding subfield (MI in this case).

**Weaknesses:**

- I do not see much utility in this work, neither from the dataset angle nor the benchmark angle. For the latter, the paper simply brings together 4 datasets that focus on face recognition, while for "benchmarks" a large collection of metrics is incorporated into the codebase without any exploration of what the metrics mean and how they should be interpreted with respect to success under model inversion. From my perspective, the proposed benchmark codebase is something you would expect in the median/ideal case from any paper that proposes a new attack/defense on model/inversion, since they would compare with previous baselines anyway. What I would have liked to see from such work would be a) a framework that makes it easy to evaluate existing attacks and defenses **across various approaches** (not just train-time defenses or GAN-based attacks), b) support for datasets beyond the narrow scope of image-based face classifiers (that seem to focus on each-person-as-a-class classification, which is far from what is actually used in face recognition systems. It may be the de-facto for MI evaluations, but this work has the chance to rectify that and target more realistic settings, or at least add support for them). A dataset paper usually proposes new datasets, but all I see in this work is a nicely packaged python repository.

- While this work claims to provide a comprehensive suite of evaluations and unified tools for model inversion, it focuses specifically on image-related datasets, and even further on attacks that require some sort of GAN-based training, and only train-time defenses. One of the core contributions of such a unified tool should be to, in fact, provide a framework that is general enough to consider most attack/defense approaches, and not just ones that are "common". For instance it might be possible that such approaches are common because other implementations are not easily available, and such a code structure only perpetuates this gap in research.

- As a general comment, please try not to shift crucial details to the Appendix. This is a dataset/benchmark paper, and relevant datasets/techniques should be properly explained in the paper. As a starting point to get more space, I think Figure 1 is unnecessarily big and adds nearly nothing to the paper. At the very least, it can be reduced by at least half.

- Since model inversion is such a visual problem in judging, especially for images, I would have liked to see visual-inspection modules in the proposed package to help the adversary/auditor better evaluate how "successful" they think their attacks/defenses are. I would also urge the authors to consider recent advances in diffusion models to consider better attack strategies. While it may not be part of published work, it can be a contribution of this work itself.

## Minor comments

-  I do not see why "dataset preprocessing" is mentioned as a module- any model training would preprocess data. This is not something unique to this benchmark.
- 'MI' is dominantly used in the privacy literature to refer to membership inference. Please consider using another abbreviation
- Barely into 2 paragraphs of the paper, there is mention of numbers (e.g. L52-56). At this point, the reader does not know what these numbers are, what they mean (compared to, say, a random adversary), or their significance. Please focus on relative trends instead of using actual numbers at a point in the paper where they do not make much sense.
- L63: "...each designated" makes it sounds like all four modules have each of the described parts- please fix the sentence.
- L66-71: what does "from diverse perspectives" mean? While I am not accusing the authors, this text in particular does read a lot like AI-generated text, and should be rewritten for clarity (and reducing fluff)
- L90: What does "...without intersecting classes" and "...the generator" mean? Please set these things up properly before diving into a very specific threat model. Similarly, Equation 1 is not clear at first- please explain it clearly and properly. As a benchmark paper, there is all the more reason for authors to clearly convey such threat models, setups, and techniques.
- Table 2: The citation for LS mentions 2023 but the reference mentions 2024- please fix this (and any similar) inconsistency.
- L275: Please clarify that FID is a measure computed across multiple samples, unlike ones like accuracy that are simply aggregated after computing sample-wise measurements.
- Appendix D: Please shift to the main paper. Something as crucial as limitations should not be buried at the very end of the paper.

**Questions:**

- How many of the attacks/defenses included in the benchmark suite already have codebases made available by the original authors, and how many are techniques that the authors of this work had to implement from scratch?

- L270: "Accuracy" here assumes a classification model, with a well-defined target class. If the model is a face-identification model that uses facial-similarity (like Siamese networks [1] or ProtoNets [2]), scores are computed based on pairwise similarity. What is "accuracy" in that case?

- Section 3.3, "Evaluation Module" - what would these metrics be for a) random guess, b) unoptimized methods that use public data for initialization?

- Section 4.1: Why are "sample diversity" metrics deferred to the Appendix?

- Table 3: What is $\delta$? Please introduce it in text first, and also explain what range of values to expect for these metrics. Without any explanation, I have no idea whether an FID value of 40 is low or high.

- Figures 2, 3 - visually, **none** of the attacks seem to be working. I cannot find a single image that matches the ground truth (in being the same person, let alone the same image). While metrics may make one thing that these attacks are highly potent, looking at these images I think model inversion is not a threat in any sense. Can the authors comment on this?

- L479: "With in-depth analysis, we have identified new insights and proposed potential solutions to alleviate them". What insights/solutions? Based on my reading all I could see is a very superficial comparison of different attacks/defenses, but no real conclusions or useful insights.

- L795: "We implemented 64x64 and 256x256 versions" - why would the attack/defense implementation change depending on the resolution, apart from a trivial change in the generator's deconvolution layer?

- L909-911: Is this protocol standard in existing works, or a contribution of this paper? If former, please refer to appropriate works. If not, please justify the deviation in data-split strategies.

### References
- [1] Liu, Weiyang, et al. "Sphereface: Deep hypersphere embedding for face recognition." CVPR, 2017
- [2] Snell, Jake, Kevin Swersky, and Richard Zemel. "Prototypical networks for few-shot learning." NeurIPS, 2017

---

### Official Review · Reviewer_k3bQ · 2024-11-03

**Soundness:** 3
**Presentation:** 2
**Contribution:** 3
**Rating:** 5
**Confidence:** 4

**Summary:**

This paper presents the first practical benchmark and a unified framework for evaluating model inversion attacks and defenses. It conducts extensive experiments with various combinations of attack and defense methods, datasets, and resolution settings etc. Through these experiments, the paper analyzes how different factors influence model inversion performance, identifying strengths and limitations of current techniques and offering valuable insights for further research in model inversion.

**Strengths:**

-	MIBench is the first comprehensive benchmark specifically designed for the MI domain, addressing a significant gap in the standardization and extensibility of MI attack and defense evaluations.
-	The benchmark’s modular design, which includes attack, defense, data preprocessing, and evaluation modules, reflects thorough engineering. This extensibility allows researchers to adapt and update the framework as new MI techniques and defenses emerge.
-	This benchmark has the potential to become a good resource in the MI field, allowing fairer evaluations of new techniques. Its reproducibility and standardization may also aid in enhancing the credibility and transparency of findings in MI research.

**Weaknesses:**

-	Writing Issues: Certain sections of the paper contain confusing terminology, such as "image reality," which would be more accurately expressed as "image fidelity" based on my understanding. There are also instances of inaccurate interpretations, such as describing WGAN-GP as a type of "GAN prior," whereas WGAN-GP is actually a training technique. These issues could potentially lead to misinterpretations.
-	Evaluation Depth: The experimental choice to limit evaluations to only 5 images per class may be insufficient, as the small sample size could undermine the reliability of the metrics. For instance, FID evaluations typically require thousands of images to produce robust results.
-	Novelty of Insights: Overall, the paper lacks significant novel insights. Some findings, such as the correlation between model predictive power and privacy leakage, have already been addressed in previous research (i.e. GMI). While this benchmark reaffirms these insights, it does not substantially deepen or expand our existing understanding in these areas.

Minor remarks:

-	L100: $\hat{z}$ should be $z^*$
-	The evaluation of LS is not included in Table 5.
-	There are minor formatting errors, such as an unclear reference to the MID in Section 2, which could be clarified. Additionally, the legend in Fig. 4(b) interferes with the visualization, and the color arrangement for defense methods in Fig. 5 is somewhat cluttered, making comparisons more challenging for readers.

**Questions:**

-  At this stage of MIA development, do you think it is still necessary to distinguish between low-resolution and high-resolution tasks as separate settings? From my perspective, high-resolution is more realistic and practical for real-world scenarios.
- In Section 4.5, why do all MI defenses fail, and why does the regularizer have no impact on the training process? This outcome appears quite counterintuitive.

---

### Official Review · Reviewer_yvsR · 2024-11-04

**Soundness:** 2
**Presentation:** 2
**Contribution:** 1
**Rating:** 3
**Confidence:** 4

**Summary:**

This work aims to unify contemporary Model Inversion Attack and Defense methods by proposing a modular toolbox called MIBench. MIBench implements a total of 12 state-of-the-art attack methods and 4 defense methods.

**Strengths:**

1) This paper is easy to follow.

2) Code is included.

**Weaknesses:**

1) **The term "benchmark" may not be applicable to this work.** Typically, benchmarks introduce new tasks/ datasets (e.g., MMLU, MMMU). In contrast, this paper proposes a toolbox that unifies existing attack and defense algorithms.

2) **The scientific and technical contributions of this manuscript are limited.**

- While the release of a general toolbox is valuable, most of the methods included already have official GitHub implementations available, which support training target classifiers, GANs, inversion attacks, and defense methods. These works also largely follow prior methods for fair comparison.

- Although the results in Tables 3 and 4 offer useful insights to the community, they are not totally new and have been discussed in previous papers. For instance, [C] has already shown that models with higher test accuracy are more susceptible to model inversion attacks.

3) **Many recent works conduct user studies to show the quality of MI attacks/ defenses.** This work should also consider incorporating human evals under metrics. Refer to [A,B] for user study frameworks.

4) Error bars/ Standard deviation for all experiments are missing.

5) Could the authors clarify whether BREP-MI and LOKT should be classified as label-only MI attacks rather than black-box MI attacks, given that neither uses soft information?

=====

[A] Nguyen, Bao-Ngoc, et al. "Label-only model inversion attacks via knowledge transfer." Advances in Neural Information Processing Systems 36 (2024).

[B] [MIRROR] An, Shengwei et al. MIRROR: Model Inversion for Deep Learning Network with High Fidelity. Proceedings of the 29th Network and Distributed System Security Symposium.

[C] Ho, Sy-Tuyen, et al. "Model Inversion Robustness: Can Transfer Learning Help?." Proceedings of the IEEE/CVF Conference on Computer Vision and Pattern Recognition. 2024.

**Questions:**

Overall I enjoyed reading this paper. But in my opinion, the weaknesses of this paper significantly outweigh the strengths.

Please see Weaknesses section above for a list of all questions.

---

### Note · Authors · 2024-11-14

I have read and agree with the venue's withdrawal policy on behalf of myself and my co-authors.